

# On the use of Convolutional Deep Learning to predict shoreline change

Eduardo Gomez-de la Peña[1], Giovanni Coco[1], Colin Whittaker[2], and Jennifer Montaño[3]

[1]School of Environment, The University of Auckland
[2]Department of Civil and Environmental Engineering, The University of Auckland
[3]Auckland Council - Air, Land, and Biodiversity Team

**Correspondence:** Eduardo Gomez-de la Peña (egom802@aucklanduni.ac.nz)

**Abstract.** The process of shoreline change is inherently complex and reliable predictions of shoreline position remain a key challenge in coastal research. Predicting shoreline evolution could potentially benefit from Deep Learning (DL), which is a recently developed and widely successful data-driven methodology. However, so far its implementation for shoreline time series data has been limited. The aim of this contribution is to investigate the potential of DL algorithms to predict interannual
shoreline position derived from camera system observations at a New Zealand study site. We investigate the application of Convolutional Neural Networks (CNNs) and hybrid CNN - Long Short-Term Memory networks. We compare our results with two established models, a shoreline equilibrium model, and a model that addresses time scales in shoreline drivers. Using a systematic search and different measures of fitness we found DL models that outperformed the reference models when simulating the variability and distribution of the observations. Overall, these results indicate that DL models have potential to
improve accuracy and reliability over current models.

## 1 Introduction

Sandy beaches — which represent a third of the world's coastlines (Luijendijk et al., 2018) — are particularly vulnerable to climate change as chronic coastal erosion is expected to occur at many locations over coming decades (Ranasinghe, 2016; Mentaschi et al., 2018). The socio-economic damage associated with coastal erosion has raised public concern (Cooper and
McKenna, 2008) and has increased awareness regarding the need for reliable coastal change projections. To fulfil this need, predictive models have been developed by the scientific community (e.g., Yates et al., 2009; Davidson et al., 2013; Vitousek et al., 2017; Montaño et al., 2021) and represent the most promising avenue towards assessing future shoreline change scenarios (e.g., Toimil et al., 2017). However, shoreline evolution is complex by nature, and hence its representation in predictive models has been a challenge undertaken with diverse modelling approaches.
One of the most traditional approaches to simulate shoreline morphodynamics is to build models that follow mass and momentum conservation laws. Models that follow these laws attempt to explicitly reproduce many of the processes involved in shoreline evolution (e.g. XBEACH, Roelvink et al., 2009). The representation of numerous processes in shoreline models has come with an inevitable high computational cost and a rapid error accumulation in long-term simulations, although some recent progress has been made in this regard (e.g., Luijendijk et al., 2019). Due to this disadvantage, process-based models are



generally deemed inappropriate for the multiple simulations required to quantify uncertainty in shoreline change predictions — an essential requirement for informed coastal management (Ranasinghe, 2020).

     Alternative models have been explored to reduce the computational costs and error accumulation found in process based models. One approach has been to use the equilibrium concept (first proposed by Miller and Dean, 2004) which assumes that a unique beach equilibrium profile will develop if a beach is exposed to unchanging wave conditions. Of the models based on the

equilibrium approach (e.g., Yates et al., 2009; Davidson and Turner, 2009), the ShoreFor model proposed by Davidson et al. (2013) includes a time-varying equilibrium condition that has allowed it to reproduce seasonal and interannual variability. However, in a relatively recent exercise of shoreline model comparison, it was found that equilibrium models struggle to reproduce faster scale oscillations (Montaño et al., 2020). To address this, new shoreline models have been developed adopting new approaches (e.g., Vitousek et al., 2017; McCarroll et al., 2021). In particular, a data-driven shoreline model that uses

the primary time-scales in shoreline change drivers (SPADS by Montaño et al., 2021) has been able to reproduce fast scale oscillations events more accurately when compared to previous approaches.

     A novel approach to simulate coastal processes is the use of machine learning, a set of computer algorithms that rely on task performance optimization. Previously limited by its high dependency on data quantity and quality, machine learning has unveiled its potential with the global increase of both computational power and data availability. In particular, much attention

has been recently drawn to artificial neural networks, usually simply called neural networks (NNs). The simplest form of a NN is called a multi-layer perceptron, which is also the most common architecture used in coastal research (Goldstein et al., 2019). The NN has an input layer, one or more hidden layers, and an ouput layer, where each layer consists of an arbitrary (experience-based) number of nodes ('neurons'). As data passes through the series of layers, the nodes identify a mapping function between inputs and outputs; this mapping is then evaluated through a cost function in an iterative process to optimize

performance. These simple NNs can only capture low-level forms of insightful information, and much work has been put in recent times to develop networks that are able to learn high-level abstractions from data. We herein refer to these deeper versions of neural networks as Deep Learning (DL) models. DL models have made major breakthroughs in the last decade when applied to computer science (e.g., Kobayashi et al., 2009; Hinton et al., 2012a), and more recently —although rapidly expanding— earth (Reichstein et al., 2019) and water sciences (see Sit et al., 2020, for a review). At the core of this Deep

Learning boom are Convolutional Neural Networks (CNNs) and Long Short-term Memory networks (LSTMs), representing the state-of-the-art architectures being actively developed (e.g., Lees et al., 2022).

     CNNs were initially introduced in their modern version by LeCun et al. (1989), and are designed to automatically extract features from raw input data. By operating on raw data, CNNs can adaptively learn from large input sequences and detect which features are the most relevant for the problem. As CNNs extract features regardless of how they occur in data, they

particularly stand out at detecting compositional hierarchies, a property where high-level features are composed of low-level ones (LeCun et al., 2015). This property is commonly found in many natural signals (Murray, 2013), including coastal systems. The ability to detect compositional hierarchies regardless of prior knowledge have made CNNs a breakthrough tool for object recognition in computer science (e.g., Krizhevsky et al., 2012), with later applications in diverse fields including coastal image classification (e.g., Buscombe and Goldstein, 2022; Buscombe et al., 2023; Ellenson et al., 2020).





For problems where sequential information and temporal dependencies are involved, LSTMs (Hochreiter and Schmidhuber, 1997) have been developed to learn long-term temporal dependencies within data. Accounting for temporal dependencies could be of special relevance for coastal processes that require longer time-scales (e.g. weekly, monthly) to be modelled, such as post-storm beach recovery (Castelle and Harley, 2020). However, in a recent modelling competition where LSTMs were implemented for shoreline prediction (Montaño et al., 2020), LSTM results showed to be competitive, although behind by $\sim$ half of the competition's models. A yet unexplored avenue for shoreline prediction is combining CNN and LSTM approaches, where this hybrid approach has shown potential to predict other earth-science phenomena (Reichstein et al., 2019), especially when the phenomenon's features evolve in time, such as forecasting precipitation (Shi et al., 2015), ocean temperature (Zhang et al., 2020), and atmospheric seasonality (Gupta et al., 2022).

In coastal science, DL models have been applied to modelling the surf-zone sandbar (e.g., Pape et al., 2007, 2010), sand ripples (e.g., Yan et al., 2008), and surf-zone sediment suspension (e.g., Yoon et al., 2013). The relative success of DL to address both high-dimensionality and nonlinear relationships has resulted in a sharp increase in the use of these methods within the scientific coastal community (see Goldstein et al., 2019, for a review). However, time series prediction in coastal science has been left out of the DL boom, as noted in the review of Goldstein et al. (2019). This research niche has recently started to be explored, with efforts including shoreline prediction (e.g. Montaño et al., 2020; Calkoen et al., 2021). In particular, the implementation of recent advances in DL — such as CNNs and CNN-LSTMs— could be beneficial to incorporate the strong auto-correlation, compositional hierarchy and memory/storage effects of shoreline evolution (Reichstein et al., 2019).

To form a better view of DL models' potential to predict shoreline time series, we investigated the performance of CNNs and CNN-LSTMs through a series of numerical experiments. We used SPADS (Montaño et al., 2021) and ShoreFor (Davidson et al., 2013) as benchmarks to compare DL results with established approaches. Section 2 (Methods) provides a detailed description of CNNs and CNN-LSTMs, along with a description of their calibration process, data used, study site, and the experimental design. Section 3 (Results) presents the findings of the numerical experiments along with the predictive capability of the DL models. Section 4 (Discussion) discusses the results of the experiments, highlighting the advantages and disadvantages of the DL approach before presenting the conclusions.

## 2 Methods

### 2.1 CNNs

Motivated by the initial success of CNNs in image classification (Krizhevsky et al., 2012), these networks have been implemented successfully in different domains (LeCun et al., 2015) including time series analysis in water sciences (e.g., Van et al., 2020). CNNs have shown to be robust against noise, and are able to extract deep features from data by applying discrete convolutions. When applied to time series analysis, a convolution can be visualised as sliding a one-dimensional filter along a time series. The filter itself can also be seen as applying a non-linear transformation to the time series. Following Ismail Fawaz et al.





(2019), a convolution centered at a time step $t$ is given by:

$$C_t = f(\omega * X_{t-l/2:t+l/2} + b) \mid \forall\, t \in [1, T] \tag{1}$$

where a univariate time series $X$ of length $T$ is convolved (dot product $*$) with a filter $\omega$ of length $l$; a bias parameter $b$ is then added before applying a non-linear function $f$ (such as the Rectified Linear Unit). The resulting univariate time series

$C$ is the filtered version of the original time series $X$. Hence, applying multiple filters on an input time series will result in a multivariate time series, where its dimensions correspond to the number of filters used. In the convolutional layers of a CNN, the same convolution $C$ will be used to find the result for all time steps $t \in [1, T]$; this enables CNNs to learn time-invariant filters. By applying multiple filters, CNNs capture multiple features of the time series.

The deep CNNs here implemented (Figure 1) take input time series and produce predictions that map to a target time series.

The CNNs consist of two convolutional layers followed by two fully connected layers (layers in which all neurons are connected to all neurons of the previous layer). In between the convolutional and fully connected layers, a sub-sampling operation is applied (max-pooling). The max-pooling operation creates a new feature vector out of the largest values of each filtered time series vector. By producing a location-invariant features vector with reduced dimensions, max-pooling layers enhance the performance of CNNs. The features vector is then passed to the fully connected layers, where non-linear transformations are

applied. In between CNNs' fully connected layers the dropout regularization method (introduced by Hinton et al., 2012b) was implemented; when dropout is applied to a layer, a percentage of neurons is disabled with the effect of making a network more robust against overfitting. Finally, the last fully connected layer takes the non-linearly transformed result of the convolutions to make a prediction. The prediction is then evaluated with its corresponding value in the target time series.

## 2.2   CNN-LSTMs

The architecture of the hybrid networks used in this study can be viewed as an array of two submodels: a CNN unit followed by a LSTM one. We first describe here LSTMs generalities and functioning. This is followed by a description of the CNN-LSTM architecture implemented in this work. LSTMs are DL models specifically designed to have memory cells ($c$), and hence have capacity for accumulating state information. In Figure 2, an acyclic graph of a LSTM is presented. For each time step $t$, the input $x_t$ is processed and the computation of a hidden value $h_t$ is updated (being initialized as a vector of zeros in the first

step). To control the memorizing process, LSTMs use "gating" mechanisms in their memory cells , where the role of each gate is to regulate the flow of information into and out of the cell. The first gate is the forget gate (introduced by Gers et al., 2000) and consists of a logistic sigmoid function $\sigma$ and a point wise multiplication operation. This gate controls which elements of the cell state vector $c_{t-1}$ will be forgotten:

$$f_t = \sigma(W_f \cdot [h_{t-1}, x_t] + b_f) \tag{2}$$

where $W_f$ is a learnable weight matrix of the hidden state $h_{t-1}$ and the input $x_t$, and $b_f$ is a learnable bias vector. The resulting vector $f_t$ has values in the range $(0, 1)$, where a value of $1$ represents a complete information retention and $0$ represents no information retention at all. In the second stage it is decided what new information should be stored in the cell state. For this





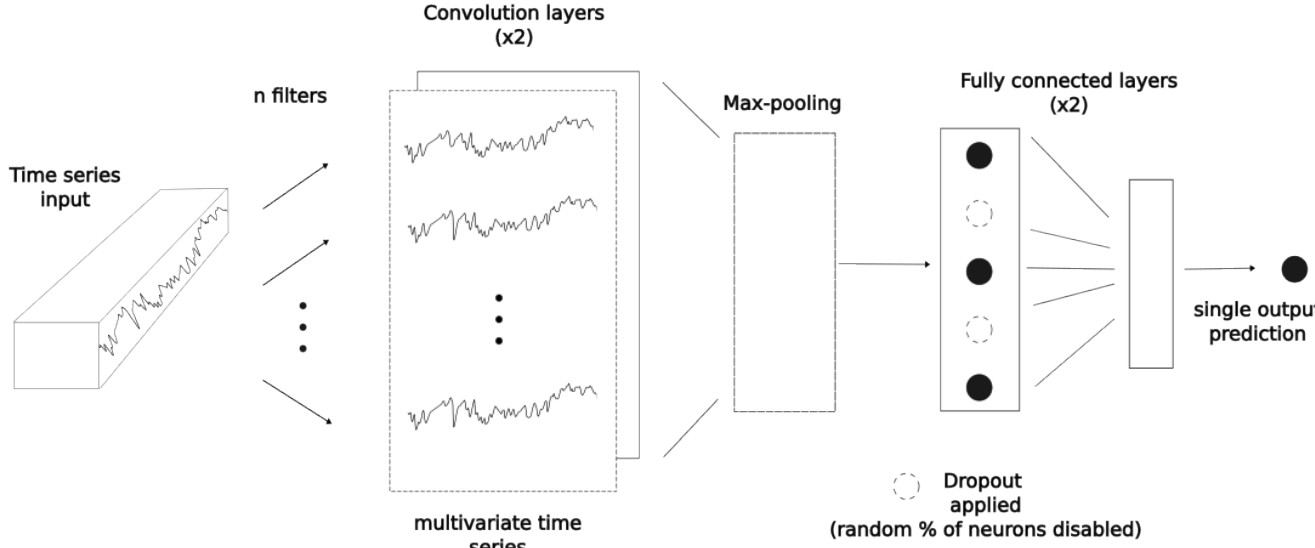

**Figure 1.** Internal operations of a CNN. A set of filters is applied to the input time series in order to produce a multivariate time series in the convolutional layers. In the Max-pooling layer a feature vector is created that is further non-linearly transformed in the fully connected layers. In the fully connected layers, Dropout is applied for regularization purposes. A single output prediction is then produced, which is directly evaluated with its corresponding value in the target time series.

aim the input gate is defined, where the values to be updated are specified:

$$i_t = \sigma(W_i \cdot [h_{t-1}, x_t] + b_i) \tag{3}$$

where, in a similar way to the forget gate, $W_i$ is the weight matrix, and $b_i$ a bias vector. The resulting vector $i_t$ contains information on which values will be updated, with values in the range $(0, 1)$. This is followed by the computation of $\tilde{c}_t$, a vector of new values — in the range $(-1, 1)$ — that could be potentially added to the cell state:

$$\tilde{c}_t = tanh(W_c \cdot [h_{t-1}, x_t + b_c) \tag{4}$$

where $tanh$ is the hyperbolic tangent function, and $W_c$ and $b_c$ are yet another pair of a learnable matrix and a bias vector,
respectively. In the next step, we combine eq. 2, 3 , and 4 to update the cell state:

$$c_t = f_t * c_{t-1} + i_t * \tilde{c}_t \tag{5}$$

where $*$ denotes element-wise multiplication. Finally, the output gate is defined. This gate controls the information that flows into the new hidden state $h_t$ and consists of a sigmoid function $\sigma$ that filters which parts of the cell state will be passed on:

$$o_t = \sigma(W_o \cdot [h_{t-1}, xt] + b_o) \tag{6}$$





Then the new hidden state $h_t$ is updated:

$$h_t = o_t * tanh(c_t) \tag{7}$$

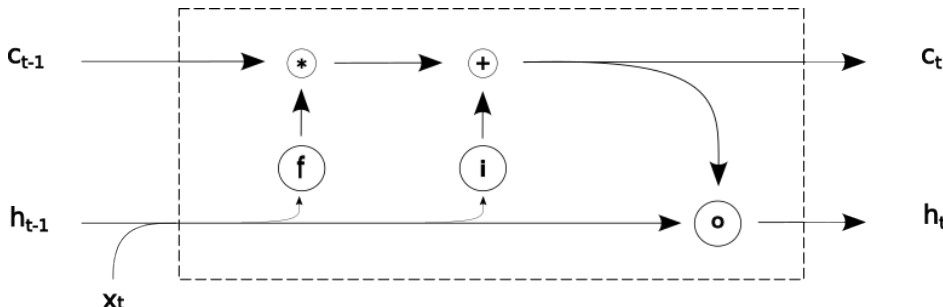

**Figure 2.** Information flow on a LSTM block, where $f$ is the forget gate (Eq. 2), $i$ is the input gate (Eq.3), $o_t$ the output gate (Eq. 6); $c_t$ and $h_t$ are the cell and hidden states, respectively, at time $t$.

The described LSTM blocks can be either stacked in parallel to create deeper networks, or used in tandem with other type of blocks to create hybrid networks, such as in CNN-LSTMs. While LSTMs were designed for sequence learning tasks (such as speech recognition), CNNs instead target spatial learning problems (such as object classification). The logic behind using

hybrid CNN-LSTM networks is that the CNN component is used to extract deep features of the input data, while the LSTM component is used to learn how those features change with time. In other words, hybrid CNN-LSTM approaches are suitable for problems that have time-evolving multi-dimensional structures, a common property of coastal phenomena, where a myriad of processes encounter at diverse temporal and spatial scales.

Similarly to previously described CNN networks, the CNN unit in the implemented CNN-LSTMs consists of two convolu-

tional layers followed by a max-pooling layer. The distilled feature vector obtained with the CNN unit is then used in the LSTM layer, where the LSTM layer allows the hybrid network to know what was predicted at the previous time step and accumulate information in the internal state (Figure 3).

### 2.3    Calibration of Deep Learning models

DL models are trained to learn the mapping between a set of inputs and a set of outputs in an iterative cycle. During the

training phase output values are calculated within the network; these values are then compared to the "true" values with a given loss function in each step of the iteration cycle. The gradient of the loss function is then calculated explicitly with respect to the network's parameters. This is called backward propagation of errors, or backpropagation for short; see Goodfellow et al. (2016) for a more detailed description. In this process, parameters and weights are adjusted to minimize the loss function in every iteration step.

Before training a network a set of parameters must be defined; these are known as hyperparameters and control the learning process of the network. In contrast to weights and biases, whose final values are derived via training, hyperparameters are



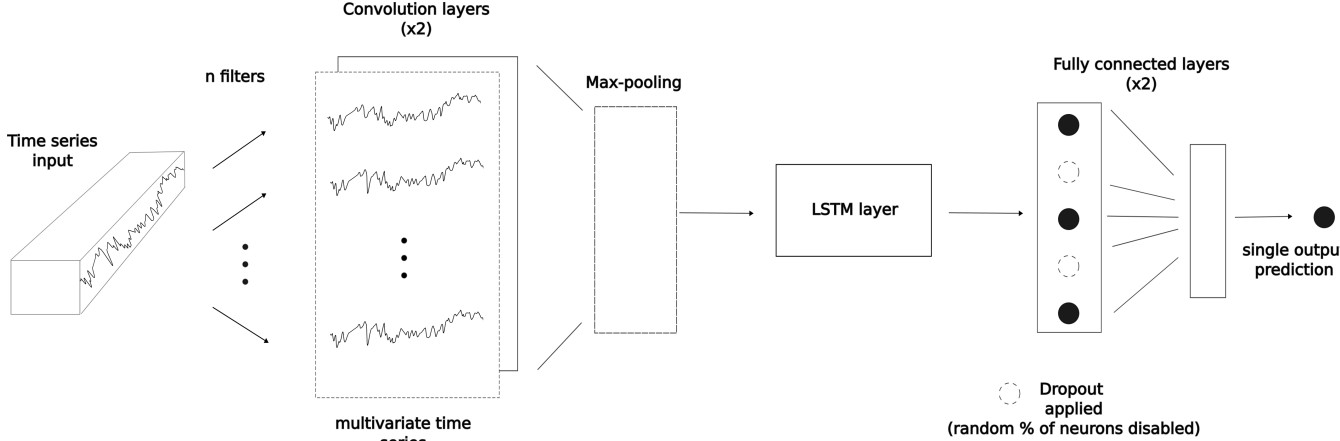

**Figure 3.** Internal operations of a CNN-LSTM. Similar operations of a CNN model (Figure 1), with the addition of a LSTM block (Figure 2) between the Max-pooling layer and the fully connected layers.

defined by the user. One example of this is the batch size, a hyperparameter that sets how many samples of the training data will be used in each iteration step before the network's parameters are updated. An epoch, defined as the period in which all the samples in the training dataset are used once to update the network, is another example of a hyperparameter.

A standard practice when calibrating DL models is to subdivide the data into three parts: the train, development, and test sets. While the train set is used for the learning process of the network — where the weights and biases values are defined — the development set is used for evaluating the DL model while tuning model hyperparameters. Once the training and tuning processes are carried out, a final unbiased evaluation is then performed on the unseen test set.

## 2.4    Data

We here describe the set of inputs and outputs used to train, develop and test the DL models. For the model inputs we used wave and atmospheric drivers, while a shoreline time series was used as the target (or output) of our models. The study site is Tairua beach, which is located in the Coromandel Peninsula, North Island of New Zealand. Tairua is a 1.2 km embayed beach with median sediment diameters ($D_{50}$) of $\sim 0.3$ mm, where the tidal range varies between 1.2 - 2 m. Shoreline position was captured with approximately daily observations over a period of 18 years (1999-2017) using a camera system at the north end

of the beach. The images obtained were georectified and processed to extract the shoreline time series — the models' target. Details of the image analysis used to obtain the shoreline time series can be found in (Blossier et al., 2016, 2017; Montaño et al., 2020). The traditional inputs for modelling shoreline position are wave bulk parameters (i.e. significant wave height $H_s$, peak period $T_p$ and direction $\theta$). We include these drivers by using the wave characteristics (at 10 m water depth) time series in Montaño et al. (2020), obtained with a SWAN nearshore wave model (Figure 4), validated with in situ measurements in 8 m

water depth. The data of the study site are freely available and have been used previously in other works (e.g., Jaramillo et al., 2021; Vitousek et al., 2021; Lim et al., 2022).




**Figure 4.** Shoreline (target) time series (a) and (b) wave bulk parameters used as model inputs ($H_s$, $T_p$, $\theta$) at Tairua.

It has been shown that sea level pressure (SLP) can drive shoreline change by influencing wave climate in some study sites (e.g., Castelle et al., 2017; Harley et al., 2010). Although atmospheric drivers and wave parameters may contain similar information, atmospheric drivers may account for mean sea level fluctuations not retained in the wave parameters (Serafin and Ruggiero, 2014; Robinet et al., 2016), which in turn could improve model results. To test at our study site, we included as model inputs the first 10 principal components (PCs) of the principal component analysis carried-out in Montaño et al. (2021) over the SLP fields in the study site's wave genesis influence area. The influence area was identified using the ESTELA (Evaluating the Source and Travel time of the wave Energy reaching a Local Area) method (Pérez et al., 2014), which has been used for wave genesis characterization in other study sites (Camus et al., 2017; Cagigal et al., 2020; Silva et al., 2020; Lu et al., 2022).



**Table 1.** Grid search hyperparameters.

| Hyperparameter | Description | Values | DL model |
|:---:|:---:|:---:|:---:|
| Training | | | |
| $b$ | Batch size | $[4, 16, 32, 60]$ | CNN, CNN-LSTM |
| Regularization | | | |
| $D$ | Dropout | $[0, 0.2, 0.4]$ | CNN, CNN-LSTM |
| Architecture-specific | | | |
| $m$ | Memory units | $[100, 200]$ | CNN-LSTM |
| $f$ | Number of filters | $[64, 72, 88]$ | CNN,CNN-LSTM |
| $k$ | Kernel size | $[2, 4]$ | CNN,CNN-LSTM |

## 2.5 Experimental design

Finding well-performing model architectures and their hyperparameters is usually a hard and time-consuming process in DL (Larochelle et al., 2009). This process can also be non-intuitive, as model performance is highly sensitive to hyperparameter tuning. To shed light on the appropriate parameter space in which to train DL models when predicting shoreline time series, we designed a grid search to assess model unknowns. The focus is on exploring CNN and CNN-LSTM architectures, since LSTMs have been already implemented for our dataset (Montaño et al., 2020). In the grid search, different hyperparameters were varied according to each architecture (see Table 1). For the CNN, the number of filters and kernels in the convolutional layers was included in the grid search, while in the CNN-LSTM filters and kernels were varied in the convolutional layers and memory blocks were varied in the LSTM layers. For all networks, the batch size, and the percentage of disabled neurons when applying Dropout were varied in the grid search.

To train and evaluate the DL models, the data split was 14 years for the train set, 2 years for the development set, and 2 years for the test set. To carry out an efficient learning process, the inputs and outputs of the training data were normalized by removing the mean and dividing by the standard deviation. In the calibration process of DL models, it is common practice to calibrate a single network with varying random seeds. We instead used fixed realizations to allow a straightforward and reproducible comparison with the benchmark results of ShoreFor (Davidson et al., 2013) and SPADS (Montaño et al., 2021) shoreline models.

Simultaneously, the designed grid search addressed the role of drivers and loss functions on network performance. With respect to the model drivers, the grid search explored wave-bulk parameters, atmospheric patterns, and the combination of these two as model inputs for DL models. These "driver" experiments were designed to gain insight into which shoreline change drivers might be relevant for the study site. In terms of the loss functions — essential components when training and evaluating DL models — the grid search addressed the use of the mean squared error (MSE) and the modified Mielke's index (Duveiller et al., 2016). We use the modified Mielke's index because it can be seen as a naturally bounded extension of the Pearson correlation coefficient it also satisfies being adimensional, and symmetric (for more see Duveiller et al., 2016).



Mielke's index has previously shown useful for evaluating the ability of shoreline models to reproduce fast-scale oscillations
(order of days) (Montaño et al., 2021) and can be written as (Duveiller et al., 2016):

$$\lambda = 1 - \frac{n^{-1}\Sigma_{i=1}^n (x_i - y_i)^2}{\sigma_x^2 + \sigma_y^2 + (\overline{x} - \overline{y})^2 + \kappa} \tag{8}$$

where

$$\kappa = \begin{cases} 0, & \text{if } r \geq 0 \\ 2\left| \sum_{i=1}^n (x_i - \overline{x})(y_i - \overline{y}) \right|, & \text{otherwise} \end{cases}$$

$n$ is the number of observed ($x$) and modelled ($y$) values, where $\overline{x}$, $\overline{y}$ are the mean values and $\sigma_x$ and $\sigma_y$ the standard deviations
of the observed and simulated values respectively. $r$ is the Pearson correlation coefficient.

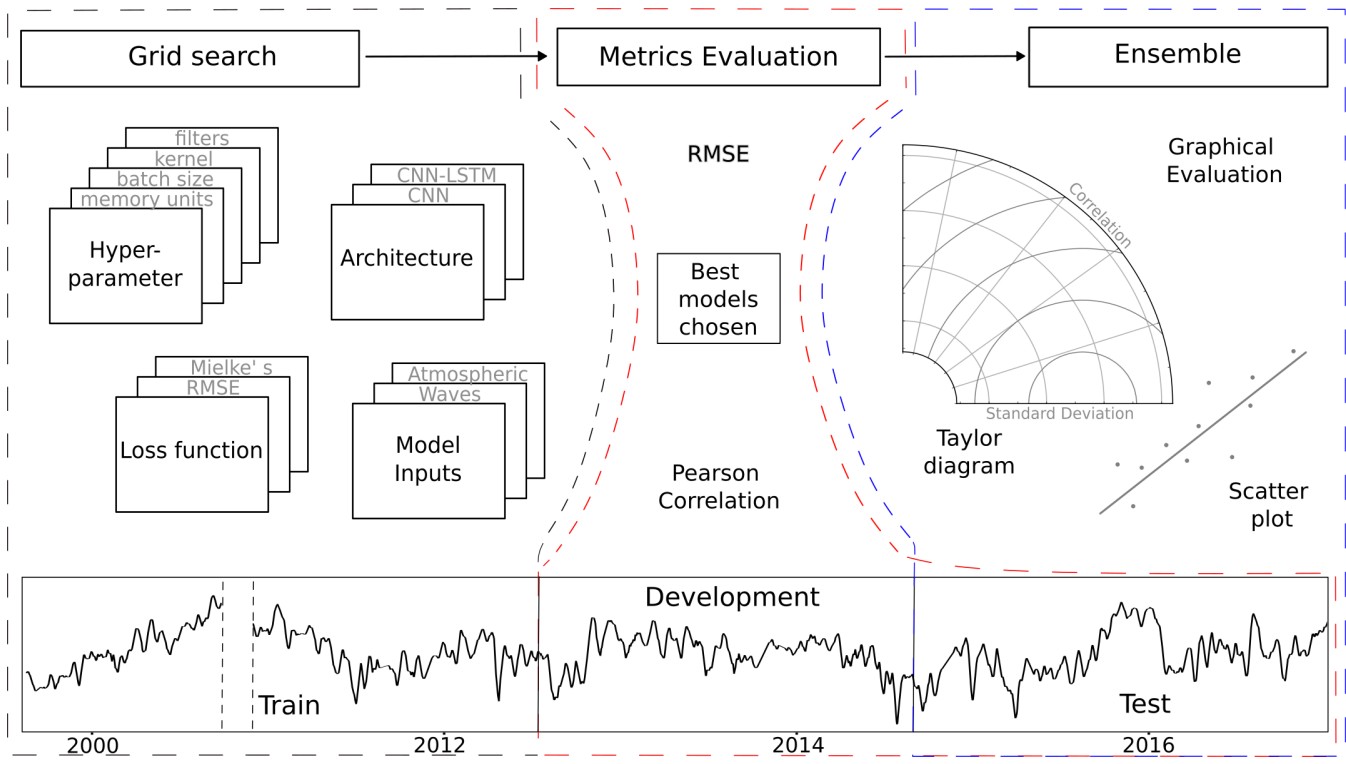

**Figure 5.** Conceptual representation of the grid search (black dashed line), metrics evaluation (red dashed line) and ensemble (blue dashed
line) processes for CNNs and CNN-LSTMs. For the grid search, the addressed model unknowns were: hyperparameter values, architectures,
loss functions, and model inputs. Once the models were trained with 14 years of the data (train set), the models were evaluated based on their
RMSE and Pearson Correlations for both the development set (2 years) and the test set (2 years). Once the best-performing models were
selected and ensembled, a further graphical evaluation was performed on the test set.

Taking into account the architectures, drivers, hyperparameters, and loss function variations, in the grid search here imple-
mented around 1000 DL models were tested. Once the experiments were carried out, we further selected the best-performing





**Table 2.** Metrics of the benchmark models on the test set.

|  | ShoreFor | SPADS |
|---|---|---|
| RMSE | 4.08 | 5.41 |
| Pearson correlation | 0.59 | 0.39 |
| Mielke's index | 0.47 | 0.32 |

models and performed averaging ensembles. The performance of the ensembles was analyzed with Taylor diagrams (Taylor, 2001) and scatter plots. A graphical summary of the variables tested within the grid search and the metrics further used to evaluate the DL models can be seen in Figure 5.

## 3 Results

We start this section by presenting the results of the driver experiments; this is followed by an analysis of loss function and fine-tuning effects on model performance. We then compare the best-performing model ensembles to the reference models, where the predictive capability of mean behavior and variability is assessed.

In order to visualize the performance of the multiple tests carried out with the grid search, we present a parallel coordinate plot (Figure 6). Each subplot in Figure 6 shows the performance of a specific set of numerical experiments, where each set is defined by the model architecture, combination of drivers, and loss function used. For all experiments, the values of the hyperparameters in Table 1 were used. An initial evaluation of model performance can also be seen in Figure 6, where a Pearson threshold of $\geq 0.6$, and a RMSE threshold of $\leq 5\,m$ were set in both the development and test sets. These thresholds allowed the detection of DL models that outperformed the benchmarks in terms of Pearson correlation, while retaining a similar RMSE when compared to the best-performing benchmark (ShoreFor), see Table 2.

### 3.1 Driver Experiments

As previously described, the grid search explored wave-bulk parameters, atmospheric patterns and the combination of these two as model inputs for the DL models. These "driver" experiments were designed to gain insight into which shoreline change drivers might be relevant for the study site and period. Of all driver experiments, DL models trained solely with atmospheric patterns (represented by SLP PCs) obtained the lowest Pearson correlation coefficients (0.1 - 0.3) and highest RMSE values ($> 6\,m$) when compared within observations on the test dataset, while higher Pearson correlation values (0.3 - 0.5) and lower RMSEs ($5 - 6\,m$) were obtained when SLP PCs along with wave-bulk parameters were used as model inputs (Figure 6). The best-performing DL models were the ones trained solely with wave-bulk parameters, where the highest Pearson correlation values ($> 0.6$) and lowest RMSEs ($4 - 6\,m$) were obtained (Figure 6).

The performance achieved with only-wave-trained DL models suggests that among the two drivers tested, only wave inputs were relevant for shoreline change prediction at the study site. Although it has been discussed in previous work by Montaño et al. (2021) that differences between waves and SLP drivers could improve model performance when used together, our results



indicate that the information contained in the SLP-PCs does not add value to DL model predictions. This can also be seen in the apparently lower model performance when both drivers are used, an effect probably due to the well-known DL models' sensitivity to unnecessary data. Although SLP PCs were found to have little influence on shoreline change for the analyzed time scale, these variables remain plausible model inputs and in fact, the importance of atmospheric controls in shoreline change is being increasingly clear (e.g., Castelle et al., 2017; Vos et al., 2023).

## 3.2 Loss function and fine-tuning effects

As described in section 2.5, the grid search also explored loss function and fine-tuning effects on previously selected architectures. The aim of the loss function trials was to explore the potential of Mielke's index for training DL models. In Figure 6 we can see that in models that used both wave parameters and SLP PCs as inputs and MSE as the loss function, the average Pearson correlation obtained from the test set was $\sim 0.33$ with maximum values of $\sim 0.55$, while the RMSE minimum was $\sim 4.6\,m$. When Mielke's loss function was used instead, the average Pearson correlation obtained on the test set was $\sim 0.43$, with maximum values of $\sim 0.63$, while the RMSE minimum was $\sim 4.3\,m$. In other words, when Mielke's index was used as the loss function, higher correlations were obtained, although RMSE values remained similar when compared to MSE-trained networks.

The aim of exploring fine-tuning effects was to ensure robust results by obtaining populations of well-performing DL models. This was done by previously hand-picking good-performing network configurations and then performing a grid search on nearby values on the parameter space. In general, DL models proved sensitive to chosen drivers and loss function, while not much sensitivity to model hyperparameters was found, as expected. In Figure 6 we can see 12 CNN models that met with the thresholds RMSE $\leq 5\,m$ and Pearson $\geq 0.6$ for both development and test sets, while 43 models that met those thresholds were obtained with the CNN-LSTM architecture.

## 3.3 Predictive capability

To compare the performance of CNNs and CNN-LSTMs with ShoreFor and SPADS, an averaging ensemble of the 10 best-performing DL models over CNN and CNN-LSTM populations was performed (Figure 7). Following the results obtained in sections 3.1 and 3.2, the ensemble was performed over the networks of the only-wave experiments trained with the Mielke's loss function. The first step taken to assess network performance was to calculate an absolute value error statistic (RMSE), along with normalized goodness-of-fit statistics (Pearson correlation and modified Mielke's index) on the test set. While the smallest RMSE was obtained with ShoreFor model ($4.08\,m$), the CNN-LSTM ensemble ($4.34\,m$) and the CNN ensemble ($4.57\,m$) closely followed, while SPADS obtained the highest value ($5.41\,m$). On the other hand, the Pearson correlation/Mielke index was higher for the CNN-LSTM ensemble ($0.61/\ 0.58$), followed by the CNN ensemble ($0.61/\ 0.55$), ShoreFor ($0.59/0.47$), and SPADS ($0.39/0.32$), see Table 3.

In addition to traditional RMSE and Pearson metrics, we further analyze model performance by presenting graphical results. A Taylor diagram (Taylor, 2001) to further investigate model performance in time series variability, and a density heat scatter plot to analyze model performance on data distribution, are presented in Figure 8. Both CNNs and CNN-LSTMs have



**Figure 6.** Parallel coordinates of the numerical experiments carried out with CNN and CNN-LSTM networks. For each experiment, the first two attributes from left to right are the Pearson correlation obtained on the test and development set. The next two attributes are the RMSE on the test and development sets. The remaining 5 attributes are the hyperparameter values tested in the grid search: Dropout percentage (D), number of filters (f), number of kernels (k), batch size (b), and memory units (m). Pink bars indicate the thresholds Pearson>0.6 and MSE<5, which indicate better or similar performance than the benchmarks. Models that simultaneously complied with these thresholds on the development and test sets are colored red.





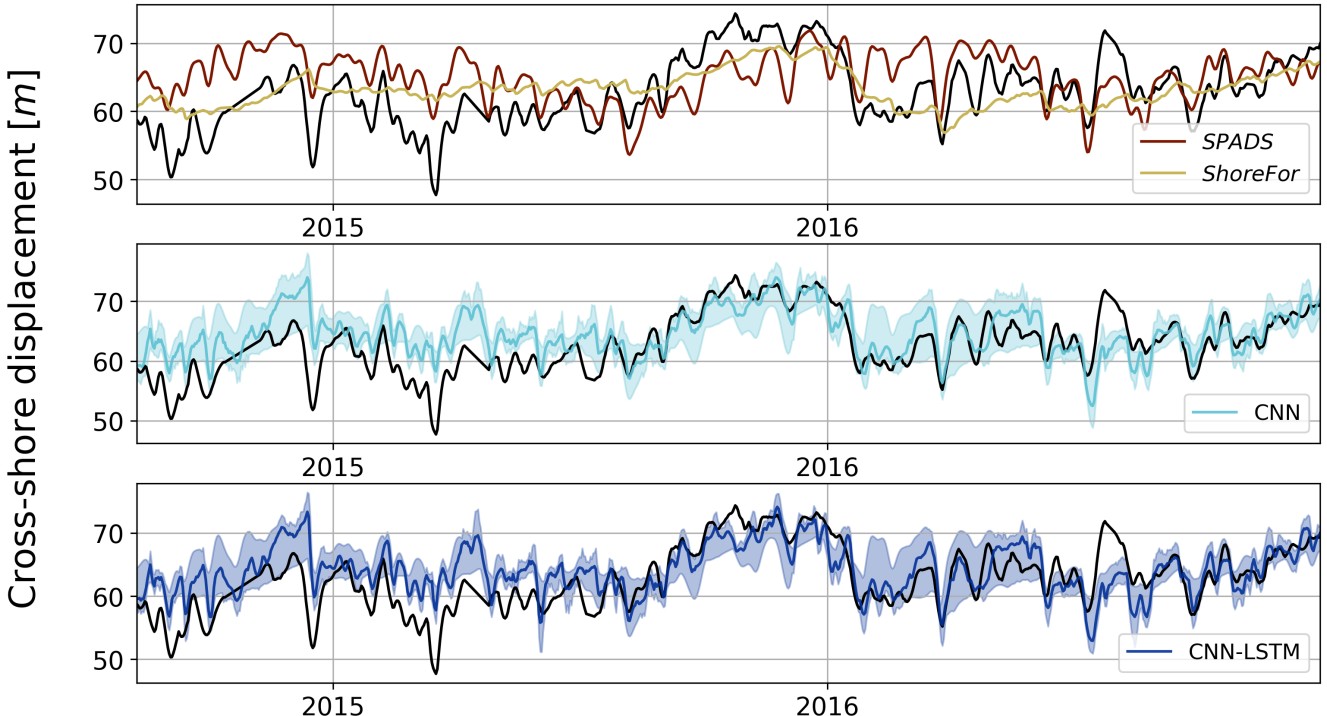

**Figure 7.** Shoreline observations (in black) and predictions for the test set using ShoreFor and SPADS (top), CNN ensemble (middle) and CNN-LSTM ensemble (bottom). For the CNN and CNN-LSTM ensembles, min-max envelope is represented by shades.

**Table 3.** Metrics of DL and reference models on the test set.

|  | ShoreFor | SPADS | CNN | CNN-LSTM |
|---|---|---|---|---|
| RMSE | 4.08 | 5.41 | 4.57 | 4.34 |
| Pearson correlation | 0.59 | 0.39 | 0.61 | 0.61 |
| Mielke's index | 0.47 | 0.32 | 0.55 | 0.58 |

the best locations in the metric space comprised by the Pearson correlation, centered RMSE, and standard deviation, with a standard deviation of $\sim 4.3\,m$ for both network populations. SPADS is the second best in terms of standard deviation ($3.5\,m$) predictability, while ShoreFor has the lowest time series standard deviation of all models ($2.5\,m$). These results suggest that, out of the analyzed models, DL models provide excellent performance as exhibited by a balance between standard deviation, Pearson correlation, and RMSE when reproducing the observations.

Our results also indicate that appropriately trained DL models were the most accurate models when reproducing the distribution of observations (see Figure 9). The location of points for each model in Figure 9 showed that ShoreFor tended to reproduce only values near the mean, while SPADS had a more frequent overestimation of true values when compared to DL





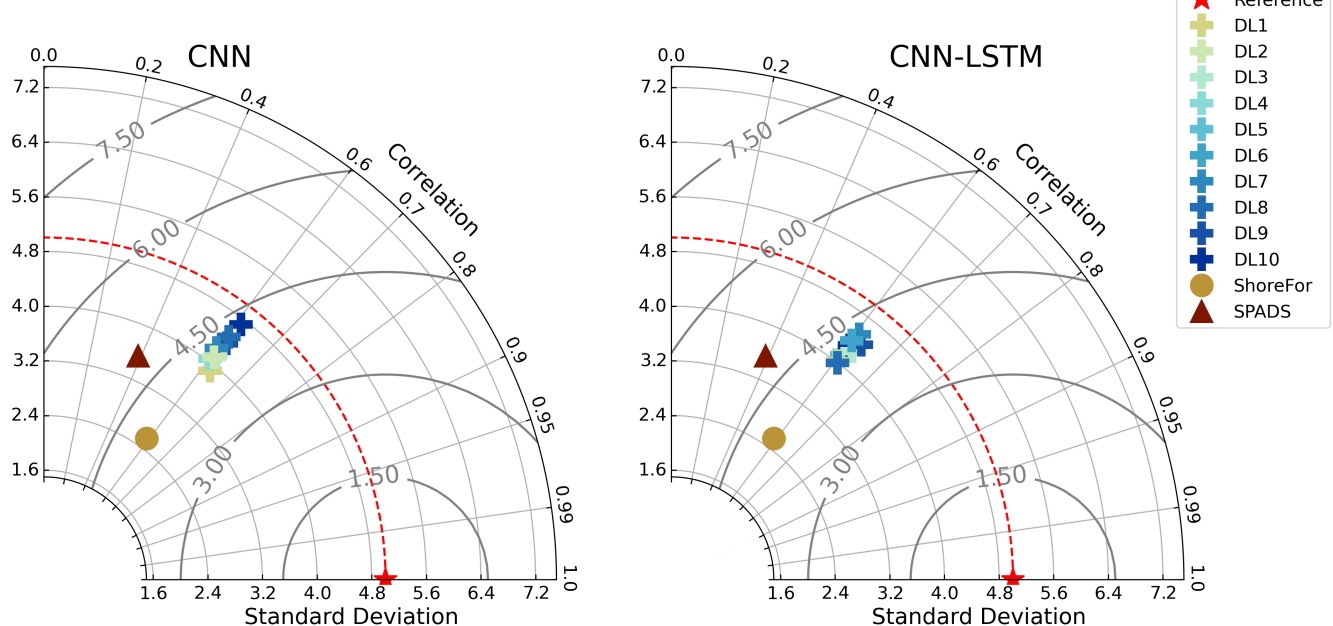

**Figure 8.** Taylor diagrams on the test period. Performance of ShoreFor (brown), SPADS (green), and individual members of the CNN ensemble (left) and CNN-LSTM ensemble (right) compared with the observations ("ideal" performance in red).

models. The absence of points in the low-value portion of all plots in Figure 9 indicates that extreme erosion prediction remains

a challenge. DL models' coverage of accretion/erosion is evident in Figure 10, where the CNN-LSTM ensemble envelope was taken as a reference and covers $53.7\%$ of the observations with no time delay, $62.1\%$ of the events allowing $\pm 1$ day shift, and $68.5\%$ of the events allowing $\pm 2$ days shift.

## 4 Discussion

While there have been previous efforts to bring Deep Learning into shoreline time series prediction (e.g., Montaño et al., 2020;

Pape et al., 2010), this work represents an extensive search and description of DL model configurations that proved effective for shoreline prediction. CNN and CNN-LSTM architectures were found effective, with the modified Mielke's function (Duveiller et al., 2016) as an appropriate loss function for the training phase. Since shoreline evolution is known to have a strong autoregressive component, and as the LSTM unit can be seen as an element that retains detailed information on previous beach states, it would be expected that the CNN-LSTM architecture would outperform the CNN, an architecture lacking a memory

unit — or an explicit memory beach parameter whatsoever. However, when comparing the performance between the CNN and CNN-LSTM, only minor differences were found (Figures 8 - 10). Since CNNs' main aim is to extract the representative features within data, one hypothesis to explain CNN's competitive performance is that —despite being traditionally depicted as



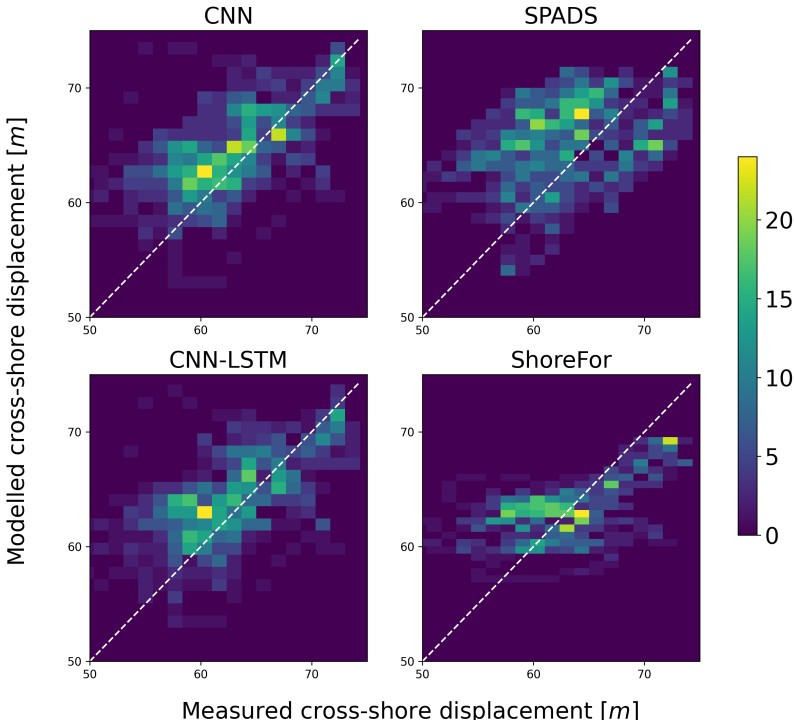

**Figure 9.** Density heat scatter plot for the shoreline predictions of DL ensembles and benchmark models on the test period. The color bar indicates the number of observations that fall within a given bin.

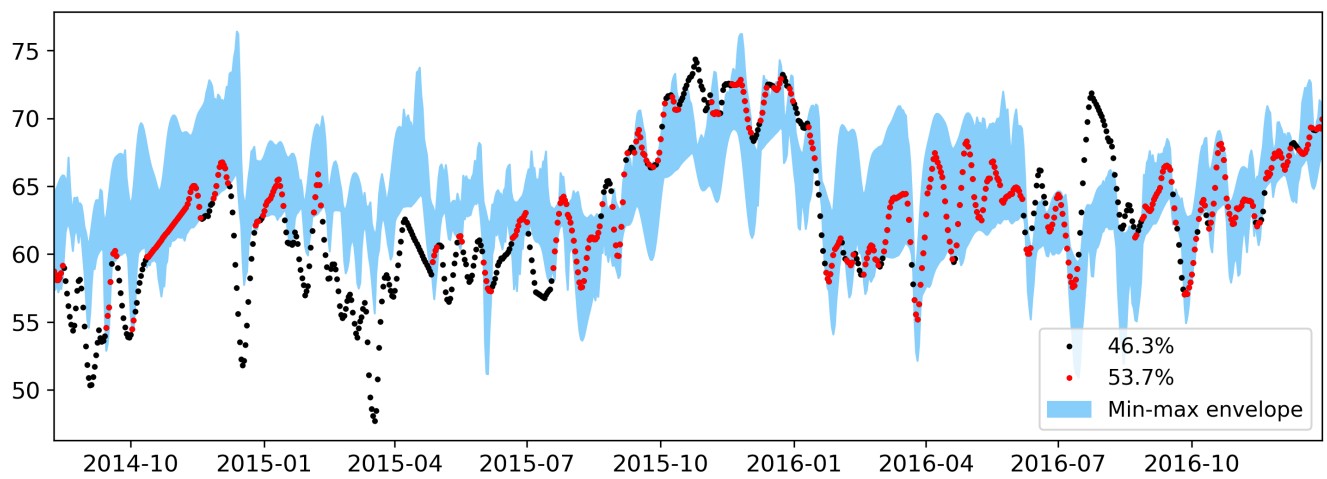

**Figure 10.** Min-max envelope for the CNN-LSTM ensemble shoreline predictions (test period). Observations that fall out/in the envelope are marked in black/red.



a strongly autoregressive process — shoreline evolution at the study site could alternatively be driven by a group of characteristics oscillations of the shoreline at different time scales. This in turn could suggest that knowing these characteristic oscillations
could be sufficient for predicting shoreline evolution, rather than only retaining detailed information on the immediate previous shoreline position. A similar result was found in (Montaño et al., 2021), where the multiscale model SPADS with no explicit beach memory parameter was used.

The prediction of shoreline change has traditionally been made with process-based and equilibrium models (e.g., Davidson et al., 2013; Vitousek et al., 2017; Antolínez et al., 2019), but even though work is attempting to adapt shoreline models to
reproduce isolated events under extreme conditions (e.g., McCall et al., 2010; Elsayed and Oumeraci, 2017) some of these models struggle to predict extreme and fast oscillations as noted in Montaño et al. (2020). As our results suggest, DL models could represent an alternative to more accurately reproduce the variability of shoreline change while simultaneously reproducing the average behavior. In particular, DL models showed to be effective when reproducing the general distribution of observations when compared to benchmarks. Despite the above, extreme events remained a challenge to predict also for the
DL approach. This could be due to the difficulty of gathering reliable data under extreme conditions (Sherwood et al., 2022), and therefore not having enough representative data of extreme events to train the models. A way this could be addressed could be by implementing a loss function in the training process that penalizes bad performance on outliers explicitly. Another possibility could be to generate synthetic data, a technique used in other fields, to compensate for the lack of data availability (e.g., Xu et al., 2020).

One of the concerns raised when using DL is its "black box"-like nature. To address this concern, an open research area in DL is the developing of models that are constrained by physics principles (e.g., Wang et al., 2022). In addition, there are techniques that bring interpretability to DL models (e.g. gradient activation maps applied for coastal studies as in Rampal et al., 2022). While new models and techniques are being developed, DL models in their current state can also aid in giving insight into the study system when compared with traditional cross-shore morphodynamic models; here we showed that DL models
trained with wave bulk parameters as their only input were the most efficient when compared to traditional models, suggesting that wave parameters were the main drivers of shoreline change for the study site and test period. On the contrary, DL models trained with atmospheric drivers did not show a competitive performance, indicating that atmospheric-related drivers might have a small contribution at the study site, as reported for New Zealand in Vos et al. (2023), albeit for longer time scales than the ones analyzed in this work.

DL is at an early stage of research in coastal science and while our understanding of these models deepens, we want to stress the importance of exploring DL through grid searching and ensemble modelling. The DL models here developed showed to be computationally efficient, enabling numerous trials that serve two purposes, first to find competitive configurations of DL models, and second to provide a guideline on how to navigate the unknowns when using DL for the first time at a study site. When implementing our models for shoreline prediction we addressed choosing among possible: (i) model inputs, (ii)
DL architectures, and (iii) model parameters. (i) A first layer of uncertainty lies on the chosen model inputs, where the use of grid searching allowed us to simultaneously test for combinations of atmospheric and wave inputs. (ii) Another layer of uncertainty is the chosen DL architecture, while previous work has been done with LSTMs in both coastal (Montaño et al.,





2020) and other water sciences areas (Kratzert et al., 2018), many architectures with potential applications for earth sciences are yet to be studied (Reichstein et al., 2019). With the implemented grid search we studied CNN and CNN-LSTMs, two types

of DL models that showed competitive performance in shoreline prediction. (iii) The third and last unknown was parameter uncertainty, where values assigned to model parameters were varied in the grid-searching process. This allowed us to run different model versions, where an average ensemble of models was found to have competitive performance. Of the works available for DL in shoreline prediction (Pape et al., 2010; Montaño et al., 2020, e.g.,), to our knowledge there is none that states the parameter values used. We here provide not only the parameter values used but also their uncertainty assessment

through ensemble modelling, which could be valuable for future works on DL models for shoreline prediction.

In previous works, diverse metrics have been used to evaluate shoreline models performance (e.g., Ibaceta et al., 2022; Jaramillo et al., 2020; Montaño et al., 2021), and although no consensus has been reached on a standard evaluation procedure, the current approach is to use a combination of absolute value error statistics, goodness-of-fitness statistics, and graphical results, as done in other water science areas (e.g., Biondi et al., 2012). For this contribution, of particular interest was the

assessment of the models' capability to reproduce the shoreline variability (which we consider to be captured by the standard deviation). Apart from an absolute value error statistic (RMSE), and a goodness-of-fitness statistic (Mielke's index), we evaluated model ability to simulate observed variability with Taylor diagrams and scatter plots. The graphic results indicate that DL models outperform traditional models when simulating the variability of shoreline behavior. The interpretation of our results could not have been achieved using only traditional metrics (i.e. RMSE, Pearson), and hence in this sense, we exhort shoreline

modelers not only to continue to use a combination of metrics and graphical results but also to assure shoreline models are being evaluated comprehensively.

## 5 Conclusions

This work investigated the potential of Convolutional Neural Networks (CNNs) and CNN- Long Short-Term Memory Networks (CNN-LSTMs) to simulate shoreline evolution derived from camera system observations. The numerical experiments allowed

us to explore the effects of varying model inputs, loss functions, and model hyperparameters in a systematic framework. The goal of this contribution was to show the potential of DL in shoreline change prediction, and hence a further exhaustive grid search for hyperparameters could be useful to find the best-performing DL model at the study site. The general findings of the experiments were:

– Absolute-value error and goodness-of-fitness statistics in conjunction with graphical results indicate that DL models

provide excellent performance when reproducing the observed shoreline position. In particular, the DL models here implemented were capable of reproducing the variability of shoreline evolution, outperforming established approaches.

– DL can aid in deriving insight into which processes drive shoreline change. In this work driver experiments indicate that shoreline change at the study site was wave-driven.

- Model performance improved when using an extended version of the Pearson correlation coefficient as the loss function,
hence demonstrating the benefit of exploring alternative metrics not only in the model evaluation stage but also in the
    training one.

Despite not having an explicit memory mechanism in their internal operations, CNNs performed similarly to CNN-LSTMs
when simulating the observed shoreline change. Since CNNs internal operations are designed to detect characteristic features
of data, we hypothesize that this could be due to the nature of shoreline evolution at the site, a phenomenon that has been
explained with oscillations at characteristic time scales in previous works.

Since DL is particularly new to shoreline time series prediction, we find value in both the grid search and model ensembles
here presented, since the former allowed for simultaneous tests on model unknowns, while the latter provided an uncertainty
assessment of model parameter values. As a closing remark, we encourage future works to make their data and model config-
urations publicly available to facilitate the implementation of DL shoreline models at multiple sites.

*Code and data availability.* Data and code of the CNNs and CNN-LSTMs models to reproduce the results here shown is available at:
https://github.com/eduardogomezdelapena/DL_shoreline_prediction.git.

*Author contributions.* EGP conceived the methodology, analyzed results, and wrote the original manuscript. G.C. conceived the study,
supervised the project, and contributed to reviewing. C.W. and J.M. contributed to reviewing.

*Competing interests.* The authors declare that they have no conflict of interest.

*Acknowledgements.* EGP is supported by The University of Auckland Doctoral Scholarsip.



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
