# Peer review of "On the use of Convolutional Deep Learning to predict shoreline change"

_EGUsphere, 2023_

## Referee Comment (RC1)

Review of manuscript Gomez de la Pena et al. "On the use of Convolutional Deep Learning to predict shoreline change" submitted to EGUsphere

**Summary and primary contribution**

This study investigates the use of Convolutional Neural Networks (CNNs) and hybrid CNN-Long-Short-Term Memory networks to predict interannual shoreline position. The target observation is a shoreline position at one location derived from 18 years of daily shoreline camera images at Tairua beach, North Island of New Zealand. The drivers include wave peak period, significant wave height and direction and sea level pressure. The results are compared with a subset of the target observation not used for training or tuning and also two models; ShoreFor and SPADS. Using a systematic search and different measures of fitness the authors conclude that CNNs models have the potential to improve accuracy and reliability over current models.

**General comments**

The combination of different metrics, graphical results (Taylor diagrams) and grid search and ensemble approach to evaluate shoreline models' performance is novel and the use of CNNs is yet not well understood, of interest of the coastal engineering community and therefore suitable for publication on the EGUsphere.

The manuscript is well structured and written in a clear fashion and well referenced.

My main concern is on the lack of some methodological important details (see also specific comments) and most critically on the rationale for constraining the prediction to a single location (not shown) to assess the shoreline predictions. This is important, as the authors indicated the oscillatory nature of the shoreline changes (L370). Should the location being close to a nodal point, the same time series of drivers will have produced virtually no changes in the cross-shore location. The ability of the model capturing the shoreline position, simultaneously at different location is not presented and the claimed improvement over ShoreFor and SPADS remains unclear.

**Specific comments**

The camera system provides images of a section of the Tairua beach but only the cross-shore position at one location has been used as a target but neither the rationale for choosing this location or a map showing the location is presented.

A short description on the set up used for the ShoreFor and SPADS model need to be included. At present, the manuscript contains very detailed information on how CNNs model has been set-up but no information is provided on the set up of the ShoreFor and SPADS models. To be consistent with authors closing remark (L373), I encourage them to make the model configuration publicly available.

Figure 4 shows both the target and drivers time series, but it is unclear if all time series have the same frequency (daily, hourly, …) and if the shoreline position was corrected for any differences in tidal elevation at the time of the camera image was captured.

---

## Referee Comment (RC2)

[referee-annotated manuscript omitted]

---

## Author Comment (AC1)

Author comments (AC) to reviewer 1 comments (RC1), manuscript Gomez de la Pena et al. "On the use of Convolutional Deep Learning to predict shoreline change" submitted to EGUsphere

Before addressing the comments, we would like to thank reviewer Andres Payo for the time invested and comments made. All comments have been incorporated and the clarity of the manuscript has certainly improved.

*RC1.1. My main concern is on the lack of some methodological important details (see also specific comments) and most critically on the rationale for constraining the prediction to a single location (not shown) to assess the shoreline predictions. This is important, as the authors indicated the oscillatory nature of the shoreline changes (L370). Should the location being close to a nodal point, the same time series of drivers will have produced virtually no changes in the cross-shore location. The ability of the model capturing the shoreline position, simultaneously at different locations is not presented and the claimed improvement over ShoreFor and SPADS remains unclear.*
*The camera system provides images of a section of the Tairua beach but only the cross-shore position at one location has been used as a target but neither the rationale for choosing this location or a map showing the location is presented.*

AC1.1: We realize this was not clear in the submitted version and that the manuscript requires some additional lines to avoid misunderstandings. Similarly to previous published works (e.g., Montaño et al. 2020, Montaño et al. 2021), we have analyzed and predicted the evolution of the average shoreline. More specifically, we have taken the alongshore-averaged cross-shore position time series as the target of our DL models. We have hence modified the manuscript to describe this in a clearer way. We notice that in Montaño et al. (2020), an international effort to model Tairua's shoreline was carried out, where 19 models of different institutions were tested on the alongshore-averaged shoreline time series of Tairua.

Lines 170-177 now read:

170    position was captured with approximately daily observations over a period of 18 years (1999-2017) using a camera system at the south end of the beach. The image analysis and tidal correction applied to the images of the camera system are in line with previous works (e.g. Guedes et al., 2011; Blossier et al., 2017; Montaño et al., 2020), where daily shoreline images with tidal levels between 0.45 and 0.55 m were selected in order to limit tidal influence. The images obtained were georectified and processed to extract shoreline time series. Then, in line with previous works (e.g. Montaño et al., 2020; Montaño et al.,

175    2021), an alongshore-averaged cross-shore position was taken in order to obtain a single time series — the DL models' target. A weekly moving average is applied to the alongshore-averaged shoreline time series to filter noise affecting the small (less than daily) time scales following Blossier et al. (2017); Montaño et al. (2021).

We have also included a new figure in section 2.4 (Data), where a map with the predominant features of the study site are highlighted:

[Figure]

Figure 4. Location of Tairua on the Coromandel peninsula in the North Island of New Zealand. Blue dots represent the installed camera system and the SWAN wave bulk parameters location.

*RC1.2. A short description on the set up used for the ShoreFor and SPADS model need to be included. At present, the manuscript contains very detailed information on how CNNs model has been set-up but no information is provided on the set up of the ShoreFor and SPADS models. To be consistent with authors closing remark (L373), I encourage them to make the model configuration publicly available.*

AC1.2: We have modified the manuscript and added a description on how the coefficients for each model are determined. We need to point out that we are not running the models Shorefor and SPADS, but only reproducing the results previously presented in Montaño et al. (2021). We have also made this last point clear in the updated version of the manuscript. The end of section 2.4 (Data) now reads:

To test the DL models, we use the time series previously presented in Montaño et al. (2021) generated with models SPADS (Montaño et al., 2021) and ShoreFor (Davidson et al., 2013); the coefficients for both SPADS and ShoreFor are determined in the models' calibration phase following optimization rules, no *a priori* information — besides wave model inputs and a shoreline target — is required.

195     The formulation of the equilibrium-based model ShoreFor (Davidson et al., 2013) used in Montaño et al. (2021) follows the modifications of Splinter et al. (2014) allowing for a general model with inter-site variability of model coefficients. The model contains two coefficients linked to wave-driven processes: (1) the memory decay parameter ($\phi$) that describes the "memory" of a beach to previous wave conditions (notice this use of the concept "memory" is different than the one used in LSTMs) and (2) the rate parameter ($c$) that describes the sediment exchange efficiency between the beach face and surf zone. At Tairua, the

200 memory parameter $\phi$ has been found to be around 220 days (Montaño et al., 2021).

    The data-driven model SPADS (Montaño et al., 2021) uses non-stationary time series decomposition methods to reconstruct shoreline oscillations at specific time-scales ($S_j$) with statistically significant driver information ($Y$). Coefficients $c$ that best fit the relation $S_j = \sum_i^N c_i Y_i$ are optimized , where $N = 1, 2...i$ correspond to the number of drivers that are significant at the time scale considered, and the subindex $j$ corresponds to the time scale of the shoreline being reconstructed.

*RC1.3. Figure 4 shows both the target and drivers time series, but it is unclear if all time series have the same frequency (daily, hourly, …) and if the shoreline position was corrected for any differences in tidal elevation at the time of the camera image was captured.*

AC1.3: The reviewer is correct and we have modified the manuscript to specify the frequency of the weekly averaged target shoreline time series, the tidal correction applied, and that the frequency of the wave time series is daily. All of these details are in line with previously published works in Tairua (e.g. Guedes et al. 2011, Blossier et al. 2017, Montaño et al. 2020, Montaño et al. 2021):

is Tairua Beach, which is located in the Coromandel Peninsula, North Island of New Zealand (Figure 4). Tairua is a 1.2 km embayed beach with median sediment diameters ($D_{50}$) of $\sim$ 0.3 mm, where the tidal range varies between 1.2 - 2 m. Shoreline
170   position was captured with approximately daily observations over a period of 18 years (1999-2017) using a camera system at the south end of the beach. The image analysis and tidal correction applied to the images of the camera system are in line with previous works (e.g. Guedes et al., 2011; Blossier et al., 2017; Montaño et al., 2020), where daily shoreline images with tidal levels between 0.45 and 0.55 m were selected in order to limit tidal influence. The images obtained were georectified and processed to extract shoreline time series. Then, in line with previous works (e.g. Montaño et al., 2020; Montaño et al.,
175   2021), an alongshore-averaged cross-shore position was taken in order to obtain a single time series — the DL models' target. A weekly moving average is applied to the alongshore-averaged shoreline time series to filter noise affecting the small (less than daily) time scales following Blossier et al. (2017); Montaño et al. (2021).

   The traditional inputs for modelling shoreline position are wave bulk parameters (i.e. significant wave height $H_s$, peak period $T_p$ and direction $\theta$). We include these drivers by using the wave characteristics (at 10 m water depth) daily-averaged time series
180   in Montaño et al. (2020), obtained with a SWAN nearshore wave model (Figure 5), validated with in situ measurements in 8

We thank once again reviewer Andres Payo for the time and comments,
The Authors

**REFERENCES**

Blossier, B., Bryan, K. R., Daly, C. J., & Winter, C. (2017). Shore and bar cross-shore migration, rotation, and breathing processes at an embayed beach. Journal of Geophysical Research: Earth Surface, 122(10), 1745-1770.

Davidson, M. A., Splinter, K. D., & Turner, I. L. (2013). A simple equilibrium model for predicting shoreline change. Coastal Engineering, 73, 191-202.

Guedes, R. M. C., Bryan, K. R., Coco, G., & Holman, R. A. (2011). The effects of tides on swash statistics on an intermediate beach. Journal of Geophysical Research: Oceans, 116(C4).

Montaño, J., Coco, G., Antolínez, J. A., Beuzen, T., Bryan, K. R., Cagigal, L., ... & Vos, K. (2020). Blind testing of shoreline evolution models. Scientific reports, 10(1), 2137.

Montaño, J., Coco, G., Cagigal, L., Mendez, F., Rueda, A., Bryan, K. R., & Harley, M. D. (2021). A multiscale approach to shoreline prediction. Geophysical Research Letters, 48(1).

---

## Author Comment (AC2)

Author comments (AC) to reviewer 2 comments (RC2), manuscript Gomez de la Pena et al. "On the use of Convolutional Deep Learning to predict shoreline change" submitted to EGUsphere

Before addressing the comments, we would like to thank reviewer 2 for the time invested and comments made. All comments have been incorporated and the clarity of the manuscript has certainly improved.

*RC2.1. While the manuscript provides sufficient details about the data and study site, it would greatly benefit from a figure that situates the reader in the study area and highlights the described elements. To enhance clarity, it is advisable to include a figure that portrays the study area's location, outlining the video camera system's position, the monitored coastline section, and the wave point utilized for forcing, among other pertinent features.*

AC2.1. We realize that a study site figure would be of great benefit, and hence we have modified the manuscript and added a figure highlighting pertinent features (Figure 4):

[Figure]

Figure 4. Location of Tairua on the Coromandel peninsula in the North Island of New Zealand. Blue dots represent the installed camera system and the SWAN wave bulk parameters location.

*RC2.2 The shoreline position time series depicted in Montaño et al. (2020) displays more fluctuations compared to the one presented in Figure 4 of this manuscript. It would be valuable to clarify whether the time series corresponds to raw data or processed data, such as a moving average.*

AC2.2. The reviewer is correct, we have modified the manuscript and clarified that we are applying a weekly moving average as done in Blossier et al. (2017), and Montaño et al. (2021), but not in Montaño et al. (2020) as the data in the above mentioned paper was provided in a raw format for the shoreline modeling competition (Shoreshop).

*RC2.3. To enhance clarity, presenting the raw data as points rather than a continuous line in Figure 4 would enable readers to identify any potential gaps in the measurements.*

AC2.3. We thank the reviewer for this comment, the figure has been modified and it does look better now:

[Figure]

Figure 5. Shoreline (target) time series, weekly averaged (a) and (b) daily wave bulk parameters used as model inputs (Hs,Tp,θ) at Tairua.

In fact, we decided to also change Figure 7 of the preprint (since it shows measurements) following the advice of the reviewer.

*C2.4. In the manuscript, it is recommended to specify the two distinct meanings of the term 'memory': one as memory cells or memory blocks in DL algorithms and the other as the 'memory decay function' employed in the ShoreFor model.*

AC2.4. The reviewer is correct and we had not realized about the potential confusion that the term "memory" could lead to. We have added a description on ShoreFor and SPADS' coefficients. We have now clarified what ShoreFor's memory decay parameter describes and stated that it is a different concept than the "memory" concept used in LSTMs. The end of section 2.4 (Data) now reads:

> To test the DL models, we use the time series previously presented in Montaño et al. (2021) generated with models SPADS (Montaño et al., 2021) and ShoreFor (Davidson et al., 2013); the coefficients for both SPADS and ShoreFor are determined in the models' calibration phase following optimization rules, no *a priori* information — besides wave model inputs and a shoreline target — is required.
>
> 195      The formulation of the equilibrium-based model ShoreFor (Davidson et al., 2013) used in Montaño et al. (2021) follows the modifications of Splinter et al. (2014) allowing for a general model with inter-site variability of model coefficients. The model contains two coefficients linked to wave-driven processes: (1) the memory decay parameter ($\phi$) that describes the "memory" of a beach to previous wave conditions (notice this use of the concept "memory" is different than the one used in LSTMs) and (2) the rate parameter ($c$) that describes the sediment exchange efficiency between the beach face and surf zone. At Tairua, the
>
> 200    memory parameter $\phi$ has been found to be around 220 days (Montaño et al., 2021).
>
>      The data-driven model SPADS (Montaño et al., 2021) uses non-stationary time series decomposition methods to reconstruct shoreline oscillations at specific time-scales ($S_j$) with statistically significant driver information ($Y$) . Coefficients $c$ that best fit the relation $S_j = \sum_i^N c_i Y_i$ are optimized , where $N = 1, 2...i$ correspond to the number of drivers that are significant at the time scale considered, and the subindex $j$ corresponds to the time scale of the shoreline being reconstructed.

*RC2.5. Has the performance of the suggested approach been assessed considering different calibration period extensions? Is there a specific minimum timeframe or minimum quantity of data necessary for the application of this methodology?*

AC2.5.  We chose a fixed calibration period to allow a straightforward and reproducible comparison with Montaño et al. (2021) results.  Although we have explored other training periods, we will describe this aspect of the calibration in a different publication that specifically addresses cross-validation.

We thank reviewer 2 for the detailed comments on typos and other minor corrections, they have all been addressed and will appear in the updated manuscript,

The Authors

**REFERENCES**

Blossier, B., Bryan, K. R., Daly, C. J., & Winter, C. (2017). Shore and bar cross‑shore migration, rotation, and breathing processes at an embayed beach. Journal of Geophysical Research: Earth Surface, 122(10), 1745-1770.

Davidson, M. A., Splinter, K. D., & Turner, I. L. (2013). A simple equilibrium model for predicting shoreline change. Coastal Engineering, 73, 191-202.

Montaño, J., Coco, G., Antolínez, J. A., Beuzen, T., Bryan, K. R., Cagigal, L., ... & Vos, K. (2020). Blind testing of shoreline evolution models. Scientific reports, 10(1), 2137.

Montaño, J., Coco, G., Cagigal, L., Mendez, F., Rueda, A., Bryan, K. R., & Harley, M. D. (2021). A multiscale approach to shoreline prediction. Geophysical Research Letters, 48(1).